# Frequency Feature Learning from Vibration Information of GIS for Mechanical Fault Detection

**DOI:** 10.3390/s19081949

**Published:** 2019-04-25

**Authors:** Yang Yuan, Suliang Ma, Jianwen Wu, Bowen Jia, Weixin Li, Xiaowu Luo

**Affiliations:** School of Automation Science and Electrical Engineering, Beihang University, Beijing 100191, China; sy1703415@buaa.edu.cn (Y.Y.); masuliang@buaa.edu.cn (S.M.); jiabowen109@126.com (B.J.); 13031041@buaa.edu.cn (W.L.); humourluo@163.com (X.L.)

**Keywords:** gas insulated switchgear, mechanical fault diagnosis, coherent coefficient, one-class support vector machine, support vector machine

## Abstract

The reliability of gas insulated switchgear (GIS) is very important for the safe operation of power systems. However, the research on potential faults of GIS is mainly focused on partial discharge, and the research on the intelligent detection technology of the mechanical state of GIS is very scarce. Based on the abnormal vibration signals generated by a GIS fault, a fault diagnosis method consisting of a frequency feature extraction method based on coherent function (CF) and a multi-layer classifier was developed in this paper. First, the Fourier transform was used to analyze the differences and consistency in the frequency spectrum of signals. Secondly, the frequency domain commonalities of the vibration signals were extracted by using CF, and the vibration characteristics were screened twice by using the correlation threshold and frequency threshold to further select the vibration features for diagnosis. Then, a multi-layer classifier composed of two one-class support vector machines (OCSVMs) and one support vector machine (SVM) was designed to classify the faults of GIS. Finally, the feasibility of the feature extraction method was verified by experiments, and compared with other classification methods, the stability and reliability of the proposed classifier were verified, which indicates that the fault diagnosis method promotes the development of an intelligent detection technology of the mechanical state in GIS.

## 1. Introduction

As a piece of control and protection equipment in power system [1,2], gas insulated switchgear (GIS) plays a significant role in high-voltage power grids. Discovering potential defects and hidden danger in the process of operation of GIS equipment in time can ensure the reliability and security of power grid operations.

The existing research on the reliability of GIS is mainly focused on the insulation fault diagnosis through signal analysis, and many experts have conducted extensive research on this topic. The main detection methods of partial discharge defects include the electrical method [3,4,5], acoustic method [6,7] and chemical method [8,9]. Aiming at the condition monitoring and diagnosis of gas insulated structures, a real-time measurement system combining signal acquisition, mode generation, feature extraction and defect recognition was proposed [10]. The ultra high frequency (UHF) method was used to analyze the characteristics of partial discharge, and short-time Fourier transform (STFT) [11] was used to describe the time-frequency characteristics [12,13]. Combined weight function classification tools and K-means clustering, and pulse parameters in both time and frequency domains were used to effectively identify noise signals and discharge pulses [14].

Compared with insulation faults, the development of intelligent diagnosis technology for mechanical faults in GIS is very slow. Under the action of electromotive force generated by AC current in conductors, the vibration signal in the fault changes correspondingly compared with the normal situation. In order to realize the intelligent diagnosis of mechanical faults in GIS, it is necessary to study in depth the characteristics of vibration signals of the GIS shell. The empirical mode decomposition (EMD) [15] method was used to analyze the vibration signal, and the characteristic matrix was defined to form the criterion of mechanical fault in GIS [16]. The full-acoustic acquisition method was used to collect different mechanical fault data, and the acoustic characteristics of signal was summarized to conduct fault diagnosis [17]. The transient vibration characteristics of GIS were analyzed by using finite element simulation software ANSYS, and the theoretical basis of mechanical defect detection technology in GIS based on vibration information was provided [18,19]. The vibration mechanism of GIS was studied in depth, and by extracting features of vibration signals using spectrum analysis, a method for detecting the mechanical state of GIS based on vibration information was proposed [20,21]. A new algorithm, which is composed of the k-nearest neighbor algorithm and the fuzzy c-means clustering algorithm, for the mechanical fault diagnosis of ultra-high voltage GIS was proposed to realize the detection of the mechanical state of GIS [22].

Generally speaking, the aforementioned documents have made great contributions to the development of mechanical fault diagnosis technology in GIS. However, due to the non-linearity, signal dispersion and noise interference of the GIS system, it is difficult to extract features and screen feature space. The features extracted from the aforementioned documents are insufficient, and the problem is more prominent when the number of samples is large. In addition, the training process of a single classifier is affected by the overall error rate, so the model may favor the majority class and ignore the minority class. The feature extraction method based on coherent function (CF) [23] can summarize the similarities of a spectrum and get the feature sets, and the union of all typical fault feature sets is selected as the feature atlas of GIS fault description. Holistic learning [24,25] is very common in machine learning [26,27]. A series of single weak classifiers are constructed and combined to classify or predict new data by a weighted or unweighted voting method. One-class support vector machines (OCSVM) [28,29,30] can solve the problem of unbalance between normal data and fault data; beyond this, it can judge the unknown faults, and the feature has been applied in the field of fault diagnosis. For example, system combining model-based diagnosis and data-driven anomaly classifiers for fault isolation used OCSVM to identify unknown faults, and the validity of the method was verified in the internal combustion engines [31,32]. SVM [33,34] can effectively divide the feature space and better classify the fault conditions.

In this paper, a new feature extraction method based on CF was proposed, and a multi-layer classifier composed of OCSVM and support vector machine (SVM) was constructed. The feasibility of the feature extraction method was verified by experiments, and the advantages of the proposed classifier were verified by comparing with the general classification methods such as Softmax [35], SVM, back propagation neural networks (BPNN) [36] and naive Bayes (NB) [37]. The main contributions of this paper can be summarized as follows:(1)GIS mechanical fault is diagnosed by a holistic approach which integrates the vibration signal acquisition system, feature extraction based on CF and a multi-level classifier composed of OCSVMs and SVM;(2)The CF is introduced into the feature screening process, and a method of feature extraction based on CF with double thresholds is proposed, which provides a new idea for feature screening and can fully describe characteristics of the vibration signal;(3)A multi-layer classifier composed of OCSVM and SVM is designed to diagnose GIS faults.

The remainder of this paper is organized as follows: Section 2 introduces the vibration information acquisition system of GIS, the experiment platform and the vibration signal analysis; Section 3 presents the method of extracting vibration features; Section 4 discusses the establishment of GIS fault classifier; Section 5 discusses the parameters of the diagnosis model and the optimization of feature space, and compares the method with other traditional diagnosis methods to prove the improvement in the diagnosis accuracy. Finally, Section 6 summarizes the contributions of the paper.

## 2. Experiments and Vibration Data Analysis

### 2.1. Experiments

In order to collect the data needed for the study of mechanical fault detection technology in GIS, a YD-81D acceleration sensor, DHF-7-3 charge amplifier and NI PCI-4472B acquisition card were used to build the vibration signal acquisition system, and the parameters of the system are shown in the Table 1.

The experimental object is a 110 kV three-phase common-box GIS experimental platform, as shown in Figure 1a. The types of faults simulated in this paper are shown in Figure 1b: (1) Isolation switch fault—the fault can be simulated by adjusting the position of the isolation switch; (2) Looseness of flange screw—the fault can be simulated by loosening three bolts used to fix flange; (3) Looseness of stone bolt—the fault can be simulated by loosening two bolts supporting GIS in section A. The experimental data recorded are shown in Table 2. The vibration data of each of the working conditions were collected 200 times with the alternating current (AC) of 50 Hz and 1000 A, forming the data set of the research in this paper.

### 2.2. Vibration Data Analysis

When AC flows into the conductor, the vibration information shows strong periodicity. The reason is that the excitation source of the GIS vibration is electromagnetic force, of which the frequency is twice that of the AC frequency [16]. Then, the traditional Fourier transform can be used to analyze the amplitude-frequency distribution characteristics of the vibration signal, as showed in Figure 2.

The main energy of the vibration signal is concentrated at the frequency around 1 kHz, and there is less energy at the frequency of 0.1 kHz. The reason is that the GIS system exhibits strong nonlinearity, and its natural frequency is about 1 kHz excited by the electromagnetic force of 100 Hz. When the mechanical fault occurs to GIS, the change of natural frequency leads to differences in spectrum under the same excitation, which are illustrated by the following remarkable differences: (1) the normal case, the isolation switch fault, the looseness of flange screw and the looseness of stone bolt have the highest frequency of vibration signal energy at 0.9 kHz, 1 kHz, 1.1 kHz and 1 kHz, respectively; (2) the energy ratio of 0.7 kHz and 1.2 kHz in fault working conditions is less than that of the normal case; (3) the energy at 2 kHz of vibration signal in flange fault is higher than that of other working conditions. In order to further analyze the frequency domain characteristics of vibration signals in GIS, three samples were taken from the vibration signals collected in each working conditions to perform a Fourier transform to obtain the spectrum, as shown in Figure 3.

It is observed that there is almost no energy distribution at the frequency point above 2500 Hz, so the spectrum range is set below 2500 Hz, as shown in Figure 3a. Due to the serious overlap of frequency points, the 700–1200 Hz frequency spectrum with more concentrated energy is further selected for observation, as shown in Figure 3b. Figure 2 and Figure 3 indicate the following conclusions: (1) The signal spectrum in the same working condition has a very high similarity, and the energy distribution at most frequency points is basically the same; (2) there is a great difference in the energy distribution at frequency points between vibration signals in different working conditions; (3) the vibration signals have energy distribution at most frequency points, and numerous frequency points with low energy cannot be ignored. The above characteristics can be used to identify the fault types of signals in different working conditions.

## 3. Feature Extraction Method of Vibration Signals

### 3.1. Principle of CF

In the field of signal processing, CF is commonly used to measure the degree of linear correlation between two signals in each frequency component. In this paper, the CF is used for feature extraction.

Suppose there are two time-domain signals *S_x_*(*t*) and *S_y_*(*t*), the calculation methods of CF are as follows [38]: (1) calculate Fourier spectrum *A_x_*(*f*) and *A_y_*(*f*) of *S_x_*(*t*) and *S_y_*(*t*), respectively; (2) calculate self-power spectral density functions *S_x_*(*f*) and *S_y_*(*f*),
(1)$$\{\begin{array}{c} S_{x}\left(f\right)=A_{x}\left(f\right){A_{x}}^{*}\left(f\right)\\ S_{y}\left(f\right)=A_{y}\left(f\right){A_{y}}^{*}\left(f\right) \end{array}$$
where *A_x_*^*^(*f*) and *A_y_*^*^(*f*) are the complex conjugation of *A_x_*(*f*) and *A_y_*(*f*), respectively; (3) calculate the cross power spectral density function,
(2)$$S_{xy}\left(f\right)=A_{y}\left(f\right){A_{x}}^{*}\left(f\right)$$
(4) calculate the CF of *S_x_*(*t*) and *S_y_*(*t*),
(3)$$C_{xy}\left(f\right)=\frac{{\left|S_{xy}(f)\right|}^{2}}{S_{x}(f)S_{y}(f)}$$

The range of *C_xy_*(*f*) is [0,1], and the larger the value of *C_xy_*(*f*_0_) at a certain frequency *f*_0_, the greater the coherence of signal *S_x_*(*t*) and *S_y_*(*t*) at the frequency of *f*_0_. *S_x_*(*t*) and *S_y_*(*t*) are irrelevant when CF is 0 and completely coherent when CF is 1. There are two advantages of the CF: (1) CF can describe the frequency commonality of two signals; (2) CF is not affected by absolute amplitude of the signals and describes the amplitude similarity of two signals at the same frequency point.

Firstly, two groups of samples are taken out from the normal signals for CF calculation, and the results are shown in Figure 4a. Then, a group of samples are extracted from the vibration signals of the normal case and the isolation switch fault respectively for coherence analysis, and the results are shown in Figure 4b.

As illustrated in Figure 4a, the waveforms of the vibration signals of the two groups of normal samples are basically the same, while the waveforms of the two groups of signals in Figure 4b are quite different (red is the normal case sample, blue is the isolation switch fault sample). The energy distribution of the two samples is consistent at most frequency points in the same working condition and has a large difference in different working conditions. The CF of two samples is calculated at the frequency points which are multiples of 10 Hz. The results show that the coherence coefficients of many frequency points are close to 1 in the same working condition, while the frequency points with coherence coefficients close to 1 in different working conditions being few in number.

### 3.2. Design Ideas of Feature Construction

Based on the coherence analysis above, the relationship between two signals at a specific frequency point can be described by the coherent coefficient between signals, thereby obtaining the relevant frequency points describing the commonality of the two signals. However, considering the dispersion of vibration signals and noise interference, it is necessary to calculate the coherence of a large number of signals of the same working condition. This paper designs a feature extraction method based on vibration information in different mechanical conditions, and the specific process is shown in Figure 5.

Step 1: Collect *m* groups of vibration signal samples of a typical type of mechanical defect, and perform a Fourier transform on each sample;

Step 2: Calculate the CF of each of two samples, set the strong correlation threshold *R_th_*, judge the coherence coefficient of the selected frequency points (multiples of 10 Hz) and the threshold *R_th_*, and define the frequency point whose coherence coefficient is larger than *R_th_* as the potential common characteristic frequency feature of this kind of working condition;

Step 3: Count the number of occurrences of each potential common frequency points, set the frequency threshold *N* = *N_th_* × $C_{m}^{2}$, where *N_th_* is the frequency threshold coefficient ($N_{th}\in \left[0,1\right]$), and define the potential common frequency points whose occurrence times are greater than the threshold *N* as the clear common frequency points of the certain working condition to build feature space.

In this paper, the sample number *m* is 200, the strong correlation threshold *R_th_* is 0.9, and the frequency threshold coefficient *N_th_* is 0.65. The coherent results of each working condition are shown in Figure 6a: (1) the characteristic frequency points above 1.5 kHz and around 0.4 kHz exist only in the flange loosening fault; (2) compared with the fault conditions, there are characteristic frequency points of about 0.5 kHz in the normal condition, but not characteristic frequency points around 0.3 kHz; (3) compared with other conditions, the condition of stone bolts loosening lacks the characteristic frequency points near 0.8 kHz; (4) only the flange loosening and the stone bolts loosening fault have characteristic frequency points near 0.6 kHz.

Figure 6b shows the characteristic frequency curves in different working conditions. The feature vector and the number of characteristic frequency points were quite different, and the union of feature points was used as the final feature to diagnose fault types.

## 4. GIS Fault Diagnosis Method Based on SVM and OCSVM

### 4.1. SVM

SVM is a method to realize the idea of structural risk minimization. The sample space is linearly partitioned by the optimal classification hyperplane. However, the problem of linearly indivisibility is often encountered in practical problems. Therefore, the data samples need to be mapped into high-dimensional feature space through non-linear transformation, so that it can be transformed into a linear separable problem. Its classification principle is shown in Figure 7.

For classification of two classes, suppose that the training set of n samples is *D* = {(*x_i_*,*y_i_*)|*i* = 1,2, ⋯,*n*}, *x_i_*∈*R^n^*, *y_i_*∈{−1,+1}, then the optimal classification hyperplane *H* [39] can be expressed as:(4)W⋅x+b=0
where *W* is the normal vector of the optimal classification hyperplane, and *b* is the constant term.

Two standard hyperplanes *H*_1_ (W⋅x+b=+1) and *H*_2_ (W⋅x+b=−1) are defined, which are planes through the samples closest to the hyperplane and parallel to the classification hyperplane. In order to maximize the classification interval of hyperplanes, the classification hyperplane is constructed by the following formula to correctly classify all samples [40]:(5)$$\{\begin{array}{l} \underset{W}{\min}\frac{{\left\|W\right\|}^{2}}{2}=\underset{W}{\min}\frac{1}{2}W^{T}W\\ s.t.y_{i}(W\cdot x_{i}+b)-1\geq 0,i=1,2,\cdot \cdot \cdot ,n \end{array}$$

Using Lagrange function to solve the above formula, the dual problem of the original problem is described as follows:(6)$$\{\begin{array}{l} \underset{a}{\max}Q(\alpha )=\sum_{i=1}^{n}\alpha_{i}-\frac{1}{2}\sum_{i=1}^{n}\sum_{j=1}^{n}\alpha_{i}\alpha_{j}y_{i}y_{j}(x_{i}^{T}x_{j})\\ s.t.\sum_{i=1}^{n}\alpha_{i}y_{i}\geq 0,\alpha_{i}\geq 0 \end{array}$$
where *α_i_*(*i*=1,2, ⋯,*n*) are Lagrange multipliers, and the maximization of *Q*(*α*) depends on the training set {*x*_i_^T^*x_j_*} and {*y_i_y_j_*}. If *α_i_*^*^ is the optimal Lagrange multiplier, the optimal hyperplane function is:(7)$$f(x)=\mathrm{sgn}(\sum_{i=1}^{n}\alpha_{i}^{*}y_{i}(x_{i}^{T}x)+b)$$
where *x* is test data.

### 4.2. OCSVM

OCSVM is an anomaly detection algorithm based on machine learning. Unlike traditional SVM, OCSVM only needs one class of samples to train an anomaly detection model, which maps training data to high-dimensional feature space through kernel function and solves an optimal hyperplane in feature space to achieve maximum separation between target data and coordinate origin, as shown in Figure 8.

The minimum objective function sought by OCSVM [41] can be described as:(8)$$\{\begin{array}{l} \min\frac{1}{2}{\left\|\omega \right\|}^{2}+\frac{1}{vl}\xi_{i}-\rho \\ s.t.f(x)=\phi (x_{i})\omega \geq \rho -\xi_{i},\xi_{i}\geq 0 \end{array}$$
where *i* is the number of training samples, *x_i_* is the sample data, *l* is the number of training samples, *ϕ* (*x_i_*) is the map of the original space to the feature space, *ꞷ* and *ρ* are the normal vectors of the hyperplane required in the feature space and compensation, respectively, the adjustable parameter *v*∈(0,1) is the upper limit of the proportion of error samples in the total sample and the relaxation variable *ξ_i_* is the degree to which some error samples are misclassified.

The Lagrange function is introduced to get the following formula:(9)$$L_{P}=\frac{1}{2}{\left\|\omega \right\|}^{2}+\frac{1}{vl}\xi_{i}-\rho -\sum_{i=1}^{l}\xi_{i}\beta_{i}-\sum_{i=1}^{l}\left(\phi (x_{i})\omega -\rho +\xi_{i}\right)\alpha_{i}$$
where *α_i_* and *β_i_* are Lagrange factors, and the dual problem is obtained by mapping the sample space to the feature space through the Gauss kernel function [41].
(10)$$\begin{array}{l} \{\begin{array}{l} K(x_{i},x_{j})=<\phi (x_{i}),\phi (x_{j})>=\exp(-g{\left\|x_{i}-x_{j}\right\|}^{2})\\ \min L=\frac{1}{2}\sum_{i=1}^{l}\sum_{j=1}^{l}\alpha_{i}\alpha_{j}K(x_{i},x_{j}),s.t.0\leq \alpha_{i}\leq \frac{1}{vl} \end{array} \end{array}$$

The analytical formula of *ρ* is obtained by solving Equation (10),
(11)$$\rho =\sum_{i=1}^{l}\alpha_{i}K(x_{i},x_{j})$$

The way to find the optimal hyperplane and get the OCSVM based anomaly detection model is represented as Equation (12):(12)$$f(x)=\mathrm{sgn}(\sum_{i=1}^{l}\alpha_{i}K(x_{i},x_{j})-\rho )$$

For training data *x*, *f*(*x*) indicates that it is positively or negatively located in the hyperplane in high-dimensional space. *f*(*x*) is a positive number and *x* belongs to a normal class; *f*(*x*) is a negative number and *x* belongs to an abnormal class. Therefore, OCSVM can identify non-target samples more accurately.

### 4.3. Fault Diagnosis Process

The mechanical fault diagnosis technology of GIS proposed in this paper includes two parts: feature extraction and fault diagnosis, and fault diagnosis is divided into state detection and fault recognition. In state detection, the first OCSVM classifier is used to distinguish normal from abnormal cases and solve the problem of unbalance between normal and fault data samples in actual detection. The second OCSVM classifier is used to distinguish known faults from unknown faults and solve the problem of misdiagnosis of unknown faults. In fault recognition, the fault types are judged by an SVM classifier. The specific diagnosis process is shown in Figure 9.

Firstly, 70% of the samples in a normal working condition are randomly selected as training set *A* for OCSVM1 training; 70% of the samples in each known type of fault are randomly selected as training set *B* for OCSVM2 training and SVM training; all remaining samples are used as test set *C* for evaluating the performance of the diagnosis model.

Secondly, the data of test set *C* is diagnosed by the OCSVM1 model. The signal with the normal test result is regarded as set *C*_1_, and the signal with the abnormal result is taken as the set *C*_2_.

Then, the *C*_2_ set is diagnosed by the OCSVM2 model. The signals which are detected as faults are regarded as set *C*_3_ and other data is taken as unknown fault set *C*_4_.

Lastly, input *C*_3_ set into the SVM diagnosis model for fault type identification.

## 5. Diagnosis Results and Analysis

### 5.1. Discussion of Parameters

*R_th_* and *N_th_* are two very important parameters in the mechanical fault diagnosis technology proposed in this paper, which jointly determine the feature space of different working conditions, and then affect the final diagnosis results.

If *R_th_* is too small, the difference in frequency energy distribution between signals cannot be effectively distinguished, and it will cause excessive characteristic frequency points, which make it difficult to select characteristic frequency points. If *R_th_* is too large, the effect of dispersion between signals is ignored.

If *N_th_* is too small, the characteristic frequency points will spread throughout the frequency domain, and the meaning of feature extraction is lost. If *N_th_* is too large, too few characteristic frequency points will reduce the accuracy of fault diagnosis.

In order to discuss the values of parameters *R_th_* and *N_th_*, this paper selected [0.5, 0.95] as the value range of *R_th_*, [0.6, 0.95] as the range of *N_th_* and 0.025 as the step size. Figure 10 shows the relationship between diagnosis accuracy and two key parameters. The diagnosis accuracy is low when both *R_th_* and *N_th_* take large or small values. When values of *R_th_* and *N_th_* are 0.65 and 0.9, respectively, the diagnosis accuracy is the highest, which reaches 98.75%.

In addition to *R_th_* and *N_th_*, the main parameters of the diagnosis model using radial basis function (RBF) [42] kernel function are listed in Table 3.

### 5.2. Diagnosis Results

After the parameters are determined, 10 experiments were conducted to test the stability of the diagnosis model, and the results are compared with the methods of Softmax, SVM, BPNN and NB. Figure 11 shows the confusion matrix obtained from the first experiment by the proposed method. The diagnosis accuracy of the normal case, isolation switch fault, looseness of flange screw and looseness of stone bolt are 100%, 100%, 97.5% and 100%, respectively. The 0 sample is diagnosed as unknown faults.

Figure 11 shows that the proposed method performs well in the first experiment. In order to compare different methods, Table 4 and Table 5 show the diagnosis accuracy and F-measure of different methods in the first experiment. Figure 12 shows the comparison of different methods in terms of accuracy and F-measure.

Table 4 and Table 5, and Figure 12 exhibit that the accuracy and F-measure of the proposed method are higher than other classification methods in each working condition, which shows that the method is more suitable than standard deep learning for feature learning in the paper. The results of 10 random experiments are shown in Figure 13.

In 10 random experiments, the diagnosis accuracy of the proposed method is above 90%, which is generally higher than Softmax, SVM, BPNN and NB. It is only slightly inferior to SVM in the Test 6 test, which indicates that the method proposed in this paper is more stable and has a better diagnosis effect.

In order to further evaluate the diagnosis model, the average and standard deviation were used as indicators to compare the diagnosis accuracy of each working condition with different methods. The statistical results are shown in Figure 14 and Table 6.

Figure 14 and Table 5 show that: (1) the average diagnosis accuracy of the proposed classifier is higher than that of other methods and the standard deviation is lower than that of other methods, which show that the proposed classifier is more stable and reliable; (2) considering the standard deviation, the diagnosis accuracy of the proposed classifier may be lower than other methods, but it is obviously superior to other methods overall, which shows that the method proposed in this paper is more effective for the classification of mechanical working conditions in GIS, and the diagnosis model is more reliable; (3) the feature extraction method proposed in this paper is effective and feasible.

To verify the ability of the diagnosis model to judge the unknown faults, the 20 groups of vibration samples of two composite faults (a combination of isolation switch fault and looseness of stone bolt, a combination of looseness of flange screw and looseness of stone bolt) were collected, respectively. Diagnosis results are shown in Table 7.

The diagnostic accuracy of the two composite faults are 85% and 95%, respectively, indicating that the diagnosis model has a good ability to identify unknown faults.

## 6. Conclusions

It is important to improve the reliability of the operation of GIS equipment and find out the potential defects in the operation in time. Thus, a holistic approach composed of a method to extract features and a multi-layer classifier was proposed in this study. First, we developed the characteristic description method based on CF. Then, based on OCSVM and SVM, a multi-layer classifier was constructed to conduct fault diagnosis.

The usefulness of feature learning was verified by a comparison among five machine learning methods in a series of experiments. The experimental results indicated that the technique of using CF for feature screening is feasible, and a new idea is provided for feature extraction. At the same time, it also proves that the classifier proposed in the paper is more stable and reliable than other methods. The fault diagnosis method proposed in this paper can play a certain role in the condition detection of GIS and promote the development of intelligent detection technology of the mechanical state in GIS.

## Figures and Tables

**Figure 1 sensors-19-01949-f001:**
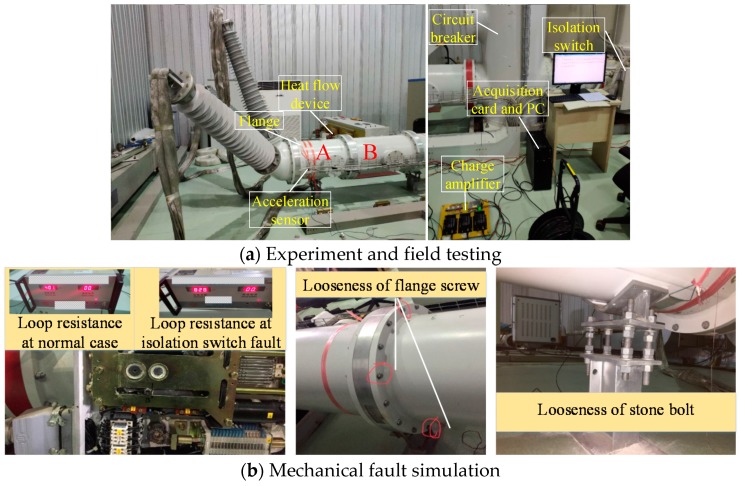
Experiment and mechanical fault simulation.

**Figure 2 sensors-19-01949-f002:**
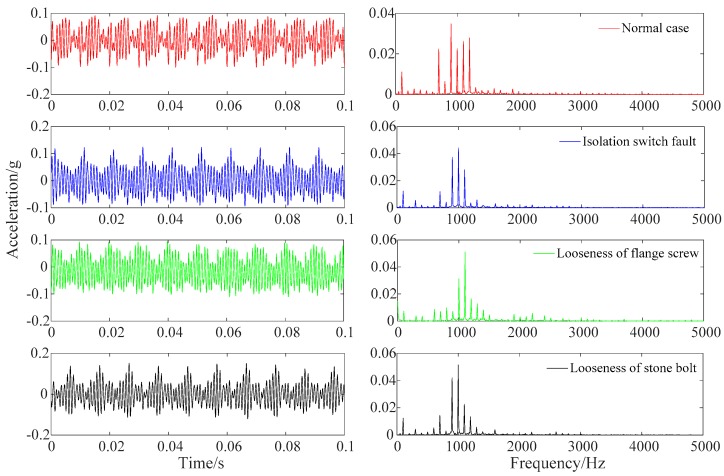
Vibration signal and its spectrum in different working conditions.

**Figure 3 sensors-19-01949-f003:**
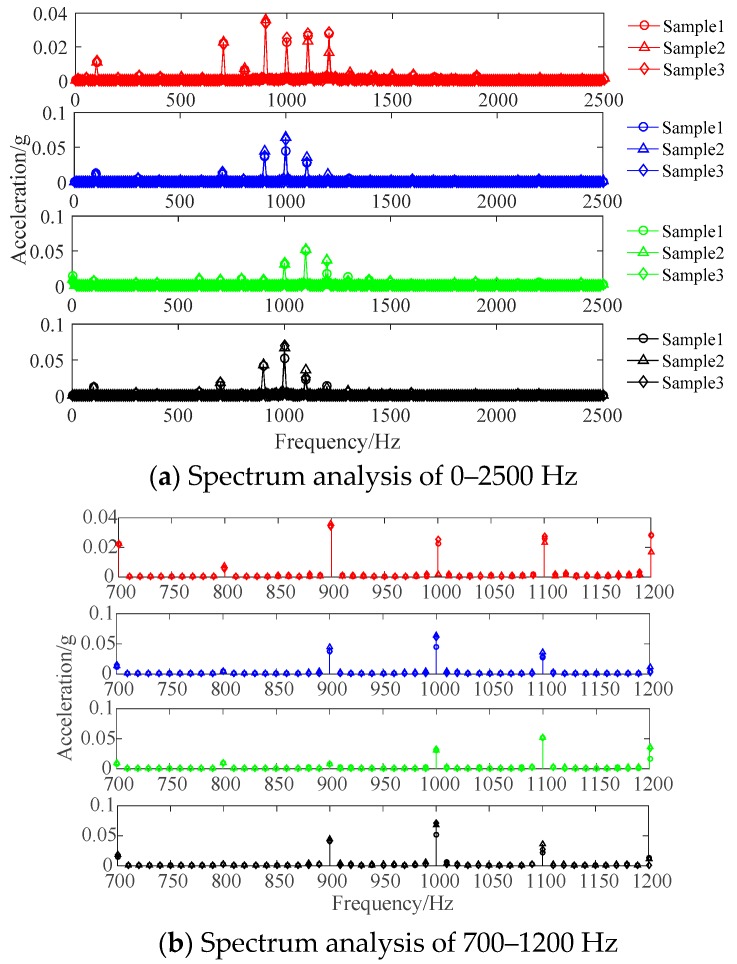
Spectrum analysis of vibration signals in different working conditions.

**Figure 4 sensors-19-01949-f004:**
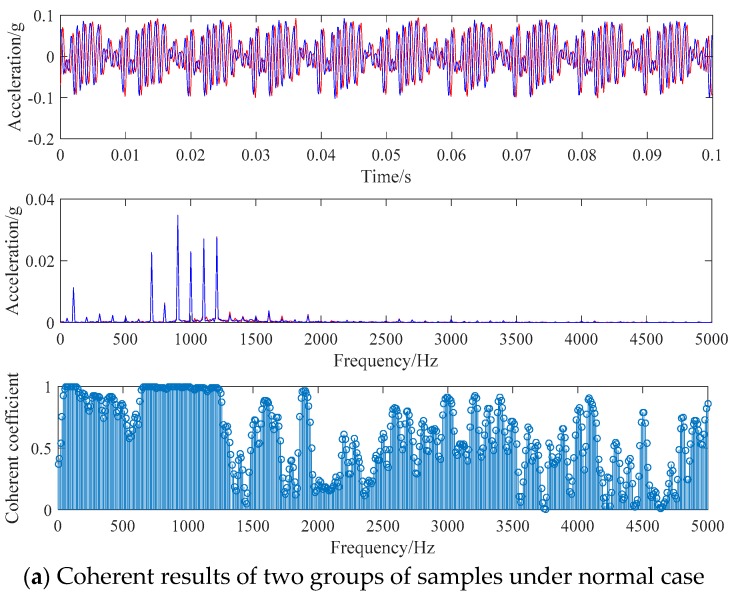
Comparison of coherent results.

**Figure 5 sensors-19-01949-f005:**
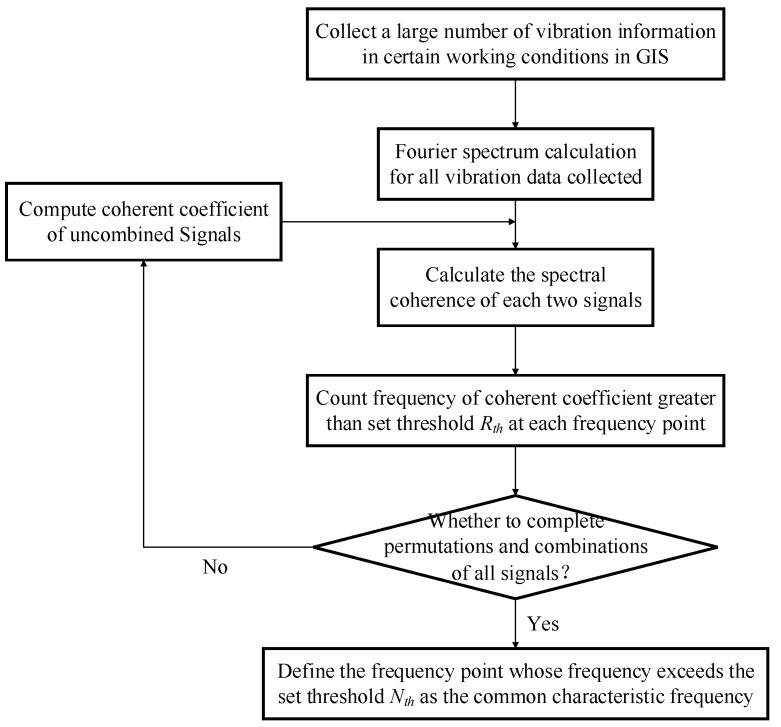
Feature extraction process.

**Figure 6 sensors-19-01949-f006:**
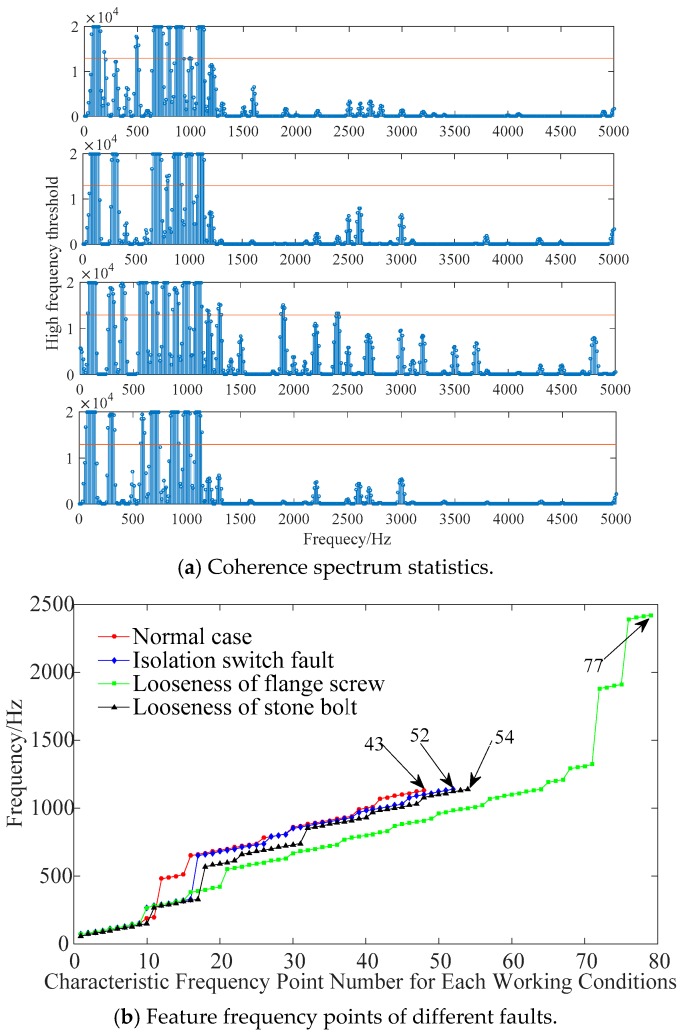
Characteristic distribution under different faults.

**Figure 7 sensors-19-01949-f007:**
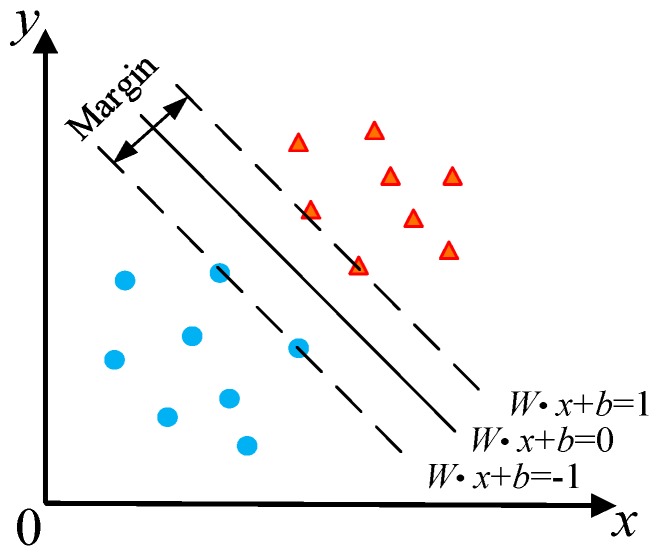
The classification principle of SVM.

**Figure 8 sensors-19-01949-f008:**
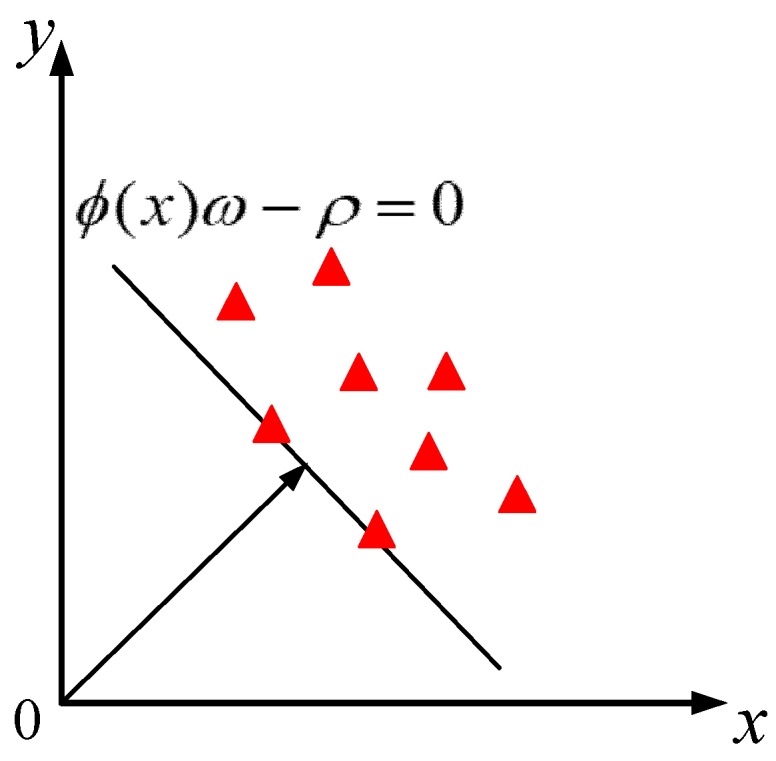
The classification principle of one-class support vector machines (OCSVM).

**Figure 9 sensors-19-01949-f009:**
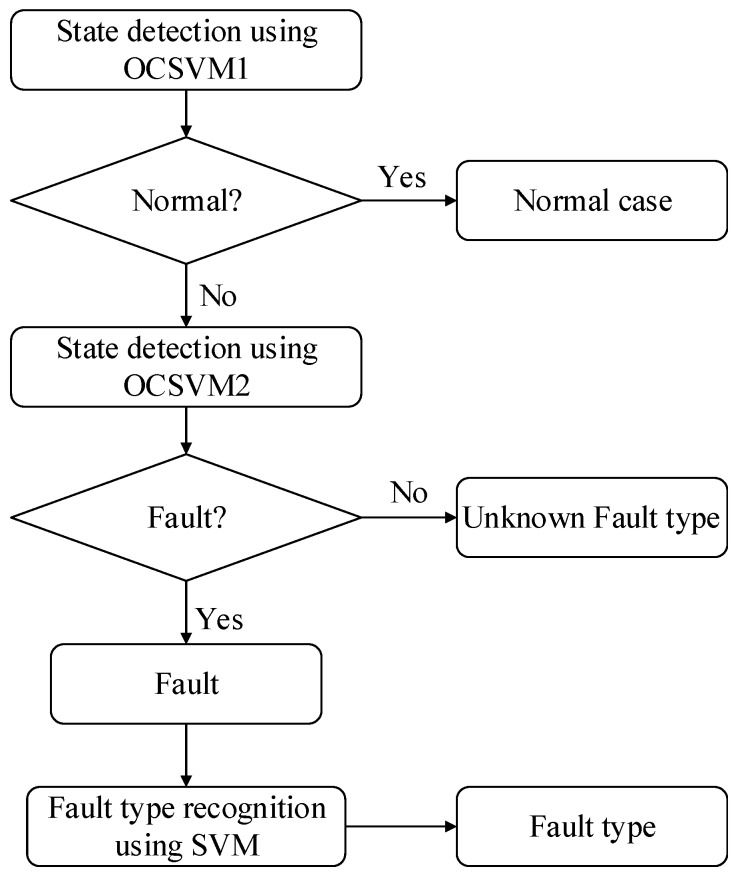
Fault diagnosis process.

**Figure 10 sensors-19-01949-f010:**
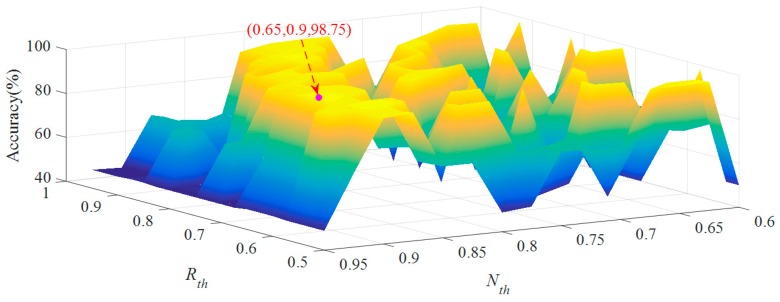
Relationship between accuracy and two key parameters.

**Figure 11 sensors-19-01949-f011:**
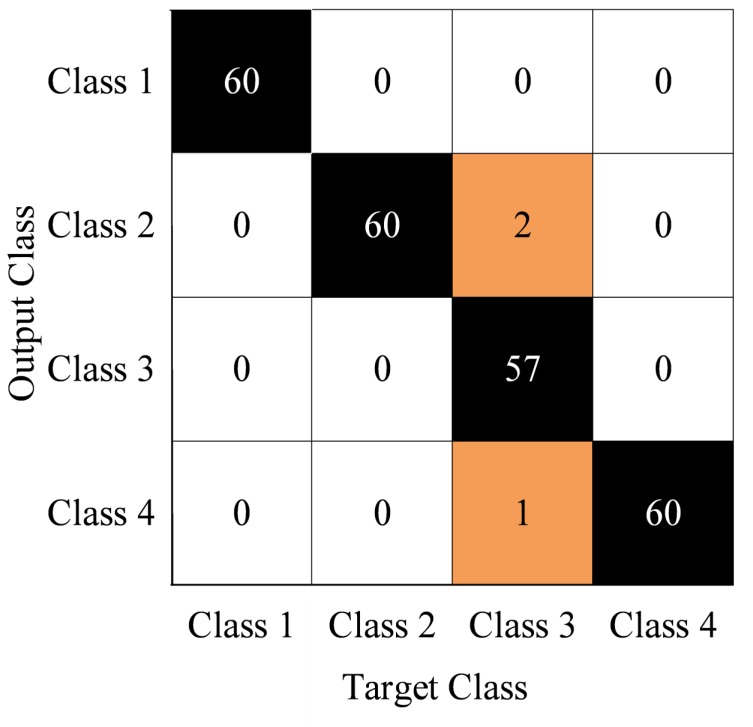
First diagnosis results (confusion matrix).

**Figure 12 sensors-19-01949-f012:**
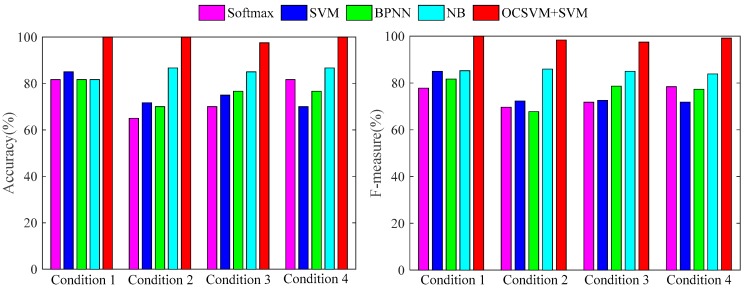
Accuracy and F-measure of first test with different methods.

**Figure 13 sensors-19-01949-f013:**
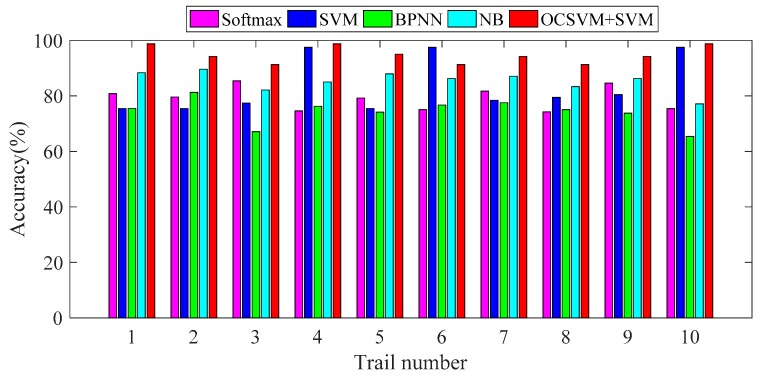
Diagnosis results of 10 tests with different methods.

**Figure 14 sensors-19-01949-f014:**
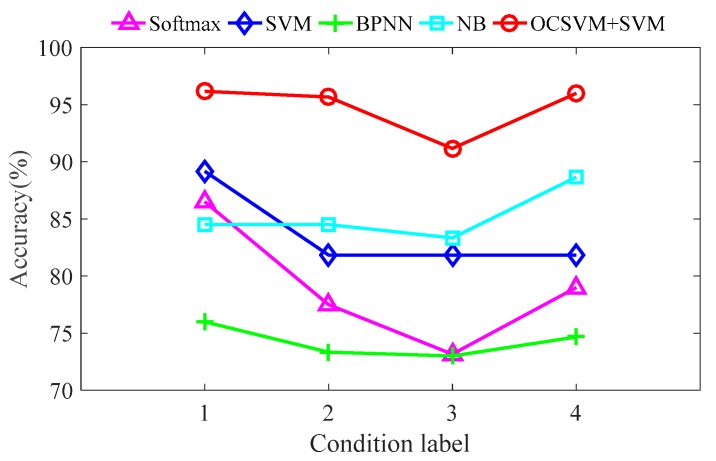
Comparison of diagnosis mean values under different working conditions.

**Table 1 sensors-19-01949-t001:** Parameters of the acquisition system.

Parameters	Value
measuring range (g)	±0.5
sensitivity (V/g)	10
maximum output voltage (V)	±5
weight of a sensor (g)	10
sampling rate (kHz)	10
sampling time length (ms)	100

**Table 2 sensors-19-01949-t002:** Summary of states of gas insulated switchgear (GIS) considered in this study.

Health Condition	Category Label	Description of State	Data Illustrate
Healthy	Class 1	Normal case	200 × 4 groups of GIS vibration data were collected under 1000 A current and four classes
False	Class 2	Isolation switch fault	
	Class 3	Looseness of flange screw	
	Class 4	Looseness of stone bolt	

**Table 3 sensors-19-01949-t003:** Parameters of the proposed method.

Description	Value
gamma of radial basis function (RBF) in OCSVM1	0.0217
nu of RBF in OCSVM1	0.66
totalSV in OCSVM1	93
rho in OCSVM1	92.3991
gamma of RBF in OCSVM2	0.02
nu of RBF in OCSVM2	0.04
totalSV in OCSVM2	17
rho in OCSVM2	16.7936
BoxConstraint in SVM (support vector machine)	0.0003
CacheSize in SVM	1000
DeltaGradientTolerance in SVM	0.001
nu of RBF in SVM	0.5

**Table 4 sensors-19-01949-t004:** Accuracy of the first experiment with different methods.

Test Method	Accuracy (%)				
	Normal Case	Isolation Switch Fault	Looseness of Flange Screw	Looseness of Stone Bolt	All Conditions
Softmax	81.667	65.000	70.000	81.667	74.583
SVM	85.000	71.667	75.000	70.000	75.417
Back propagation neural networks (BPNN)	81.667	70.000	76.667	76.667	76.250
Naive Bayes (NB)	81.667	86.667	85.000	86.667	85.000
OCSVM+SVM	100.000	100.000	97.500	100.000	98.75

**Table 5 sensors-19-01949-t005:** F-measure of first experiment with different methods.

Test Method	F-measure (%)				
	Normal Case	Isolation Switch Fault	Looseness of Flange Screw	Looseness of Stone Bolt	All Conditions
Softmax	77.778	69.643	71.765	78.400	74.396
SVM	85.000	72.269	72.581	71.795	75.411
BPNN	81.667	67.742	78.632	77.311	76.338
NB	85.217	85.950	85.000	83.871	85.010
OCSVM+SVM	100.000	98.360	97.436	99.174	98.742

**Table 6 sensors-19-01949-t006:** Diagnosis results (average and standard deviation).

Test Method	Mean and Standard Deviation of Accuracy (%)				
	Normal Case	Isolation Switch Fault	Looseness of Flange Screw	Looseness of Stone Bolt	All Conditions
Softmax	86.500 ± 6.007	77.500 ± 6.538	73.167 ± 5.119	79.000 ± 8.285	79.046 ± 4.147
SVM	89.167 ± 6.249	81.833 ± 10.045	81.833 ± 11.988	81.833 ± 13.259	83.444 ± 9.847
BPNN	76.000 ± 5.784	73.333 ± 6.894	73.000 ± 4.360	74.667 ± 7.106	74.254 ± 4.727
NB	84.500 ± 5.215	84.500 ± 3.853	83.333 ± 3.514	88.667 ± 4.216	85.290 ± 3.670
OCSVM+SVM	96.167 ± 4.648	95.667 ± 2.808	91.167 ± 2.727	96.000 ± 3.443	94.751 ± 3.088

**Table 7 sensors-19-01949-t007:** Diagnosis results of two composite faults.

Actual Working Condition	Diagnosis Result				
	Normal Case	Isolation Switch Fault	Looseness of Flange Screw	Looseness of Stone Bolt	Unknown Fault Type
Isolation switch fault and looseness of stone bolt	0	2	0	1	17
Looseness of flange screw and looseness of stone bolt	0	0	1	0	19

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
