# Peer review of "Frequency Feature Learning from Vibration Information of GIS for Mechanical Fault Detection"

_sensors, 2019, doi:10.3390/s19081949_

Reviewer 1 Report

I suggest a complete review of this text, mainly, considering the following points:

1.    English Review: e.g., “methed” etc.;

1.    Place the references according to the norms of the Sensors Journal. I observe that there is not uniformization;

2.    Do not use "et al." in references, unless it is the norm of the Sensors Journal;

3.    Use, where possible, benchmark publications. For example, reference [A] to the SVM method. Also, OCSVM [B] etc.;

4.    P.J. Werbos is the original Author of the backpropagation algorithm [C];

5.    Reference [33] for Bayes Neural Networks?;

6.    It is missing citation in important (many) parts of the text, for example, many equations (mainly), acronym “RBF” etc.,

7.    Put (it is missing) the italic style in some variables / parameters in the text;

8.    Figure 5: replace “Whether to complete permutations and combinations of all signals” by “Whether to complete permutations and combinations of all signals?”, i.e., it is missing the question mark “?”;

9.    Table II: review the information in column “Description”;

10.  In the comparative study presented were the benchmark techniques available in the specialized literature considered?

11.  I suggest to the Authors to highlight (objectively) the innovation of this proposal in relation to literature.

References

[A]  Vapnik V.N. “The Nature of Statistical Learning Theory”, 1995, Springer, New York.

[B]  Schölkopf, B.; Platt, J.C., Shawe-Taylor, J.;  Smola, A.J. and Williamson, R.C. “Estimating the Support of a High-Dimensional Distribution”, Neural Computation, 2001; Vol. 13, No. 7, pp. 1443–1471.

[C]  Werbos, P.J. “Beyond Regression: New Tools For Prediction And Analysis in The Behavioral Sciences”.  PhD. Thesis - Harvard University, 1974.

Author Response

Thank you for your letter and for the reviewers’ comments on our manuscript titled “Frequency Feature Learning from Vibration Information of GIS for Mechanical Fault Detection” (ID: sensors-470115).

Your comments have been very helpful for revising and improving our paper. They are also important and valuable guidelines significant to our research. We have studied the comments carefully and have made the relevant corrections, which we hope meet with your approval. The revised portions are marked in red in the manuscript(revision version v1). The main corrections in the paper and our responses to the reviewers’ comments are provided below:

Responses to the reviewer’s comments:
Reviewer #1: 
1. Comment:

“1.  English Review: e.g., “methed” etc.”
Response:

Thanks for your comments. We have carefully reviewed the whole paper, proofread and revised some words and grammar. The revised portions are marked in red in the manuscript.

2. Comment:

“Place the references according to the norms of the Sensors Journal. I observe that there is not uniformization.”
Response:

We have modified the format of the paper's references according to the reference format of Sensors Journal. For example, we have replaced reference “Jo, H.E., Wang, G.M., Kim, S.J., Kil, G.S. (2015). Comparison of partial discharge characteristics in SF6 gas under AC and DC. Trans. Electr. Electron. Mater., 16(6), 323–327” with “Jo, H.E.; Wang, G.M.; Kim, S.J.; Kil, G.S. Comparison of partial discharge characteristics in SF6 gas under AC and DC. Trans. Electr. Electron. Mater. 2015, 16, 323–327”.

3. Comment:

“Do not use "et al." in references, unless it is the norm of the Sensors Journal.”
Response:

Thanks for Reviewer’s guidance. We have modified the references which included "et al." and added the names of other authors.

4. Comment:

“Use, where possible, benchmark publications. For example, reference [A] to the SVM method. Also, OCSVM [B] etc.”
Response:

Thank you for the input. We have made correction according to the Reviewer’s comments and added the references to SVM and OCSVM with benchmark publications “Vapnik V.N. “The Nature of Statistical Learning Theory”, 1995, Springer, New York” and ”Schölkopf, B.; Platt, J.C., Shawe-Taylor, J.;  Smola, A.J. and Williamson, R.C. “Estimating the Support of a High-Dimensional Distribution”, Neural Computation, 2001; Vol. 13, No. 7, pp. 1443–1471”.

5. Comment:

“P.J. Werbos is the original Author of the backpropagation algorithm [C].”
Response:

Considering the Reviewer’s suggestion, we have reference “Werbos, P.J. Beyond Regression: New Tools For Prediction And Analysis in The Behavioral Sciences. PhD. Thesis-Harvard University, 1974” to Back Propagation Neural Networks(BPNN).

6. Comment:

“Reference [33] for Bayes Neural Networks?”
Response:

I carefully read the Reference [33], which proposed a Naive Bayes bearing fault diagnosis method based on enhanced independence of data. Naive Bayes, which is used in the article, is different from Bayes Neural Networks. We are very sorry for the misunderstanding resulted from our unclear description, and we made changes “Naive Bayes(NB)” elsewhere in the article.

7. Comment:

“It is missing citation in important (many) parts of the text, for example, many equations (mainly), “RBF” etc.”
Response:

According to the comments of reviewers, we have added corresponding references, mainly including equation of coherent function, SVM and OCSVM , acronym STFT、EMD and RBF.

8. Comment:

“Put (it is missing) the italic style in some variables / parameters in the text.”
Response:

Thank you for the input. After reading the paper carefully, we checked and modified the format of the full-text symbolic variable. For example, we put the italic style in parameter ϕ in the formula, picture and text, which was Ф before modification.

9. Comment:

“Figure 5: replace “Whether to complete permutations and combinations of all signals” by “Whether to complete permutations and combinations of all signals?”, i.e., it is missing the question mark “?”.”
Response:

It is really true as Reviewer suggested that question marks should be used for judgment in flowcharts and we added the missing question mark.

10. Comment:

“Table II: review the information in column “Description”.”
Response:

Because we added a table about the collection system parameters earlier, the table becomes table Ⅲ. After reviewing the information in column “Description” of Table Ⅲ, the statements of “tatalSV in OCSVM1” and “tatalSV in OCSVM2” were corrected as “totalSV in OCSVM1” and “totalSV in OCSVM2”. We also removed the "r" in front of gamma and removed the "gamma" out of the parentheses to bring it into line with the other variables.

11. Comment:

“In the comparative study presented were the benchmark techniques available in the specialized literature considered?.
Response:

In this paper, the proposed method is compared with other methods such as Softmax, SVM, BPNN and NB, which have achieved good results in the existing literature.

Literature “A New Transfer Learning Method and Its Application on Rotating Machine Fault Diagnosis Under Variant Working Conditions” used softmax to diagnose Effective data-driven rotating machine fault, and the diagnostic accuracy reached 99.05%. Literature “An Intelligent Fault Diagnosis Method Using Unsupervised Feature Learning Towards Mechanical Big Data” used softmax to process mechanical big data, and the diagnostic accuracy reached 99%.

As for the application of support vector machine(SVM), literature “A New Bearing Fault Diagnosis Method Based on Fine-to-Coarse Multiscale Permutation Entropy, Laplacian Score and SVM” used SVM to diagnose rolling bearing fault, and the diagnostic accuracy can reach 100% even.

Back Propagation Neural Networks(BPNN) was used by literature “Steam Turbine Fault Diagnosis Method Based on Rough Set with the Backpropagation Neural Network” to diagnose steam turbine fault, and the diagnostic accuracy can reach 99.57%.

Naive Bayes(NB) was used by literature “Transformer Fault Diagnosis Based on Naive Bayesian Classifier and SVR” to process data of transformer fault, and the diagnostic accuracy reached 97.22%.

As mentioned above, these techniques have been applied in the field of fault diagnosis and achieved good results, proving their feasibility. Therefore, this paper makes a comparison with these methods.

12. Comment:

“I suggest to the Authors to highlight (objectively) the innovation of this proposal in relation to literature.”
Response:

According to the comments of reviewers, we made some modifications to the article. Line 81-82,the statements of “A method of feature extraction based on CF with double thresholds is proposed, which can fully describe characteristics of the vibration signal” was modified as “The CF is introduced into the feature screening process, and a method of feature extraction based on CF with double thresholds is proposed, which provides a new idea for feature screening and can fully describes characteristics of the vibration signal” in line 81-83.

Line 365-366,the statements of “Experimental results indicated that the method to extract features is feasible, and the classifier is more stable and reliable than other methods” was modified as “The experimental results indicated that the technique of using CF for feature screening is feasible and a new idea is provided for feature extraction. At the same time, it also proves that the classifier proposed in the paper is more stable and reliable than other methods” in line 368-370.

We earnestly appreciate the reviewers’ hard work, and hope that the revised manuscript meets with your approval. We have tried our best to improve the manuscript. The changes made will not influence the content or framework of the paper.

Once again, we thank you for your comments and suggestions.

Reviewer 2 Report

Dear Author

I have read your submission entitled “Frequency Feature Learning from Vibration nformation of GIS for Mechanical Fault Detection”. The reported work is interesting. However, I have a few comments of which I hope will improve the paper.

— It might ease readability if the signal acquisition system described in Lines 94–95 are tabulated in a table.

— Line 96: It is understood that 10 kHz is equal to 100 ms, hence the second half of the sentence is redundant.

— The data was collected 200 times for each class, but there is no mention on how different is each run from the other. It would be helpful if there are plots to illustrate the distribution of noise/nonlinearities/disturbances that might make each run unique, since the manuscript also stresses on the strong presence of nonlinearities in the GIS system. 

— There is no indication on the intensity/details of the faults induced, i.e. faults with large or small magnitudes, and how many screws and nuts have to be loosened in order for the fault to be significant? 

— Please rectify Equation (7).

— Is that a symbol of a bird appearing before $p$ on Line 253?

— Line 302: Any reason behind the choice of the values?

— Since the manuscript touches on 1-SVM and the detection of unknown faults, it would also be good to compare to recently published reports that address similar issues:

— “Combining model-based diagnosis and data-driven anomaly classifiers for fault isolation”, Control Engineering Practice, 2018.

— “A combined diagnosis system design using model-based and data-driven methods”, 2016 3rd Conference on Control and Fault-Tolerant Systems (SysTol), 2016.

Lastly, there are some grammatical errors throughout the manuscript. Please revise.

Good luck!

Author Response

Thank you for your letter and for the reviewers’ comments on our manuscript titled “Frequency Feature Learning from Vibration Information of GIS for Mechanical Fault Detection” (ID: sensors-470115).

Your comments have been very helpful for revising and improving our paper. They are also important and valuable guidelines significant to our research. We have studied the comments carefully and have made the relevant corrections, which we hope meet with your approval. The revised portions are marked in red in the manuscript(revision version v1). The main corrections in the paper and our responses to the reviewers’ comments are provided below:

Responses to the reviewer’s comments:
Reviewer #2: 
1. Comment:

“1.  It might ease readability if the signal acquisition system described in Lines 94–95 are tabulated in a table.”
Response:

It is really true as Reviewer suggested that tabulating the signal acquisition system in a table eased readability, and the serial numbers of the subsequent tables are increased accordingly

2. Comment:

“Line 96: It is understood that 10 kHz is equal to 100 ms, hence the second half of the sentence is redundant.”
Response:

We are very sorry for our negligence of the statement. We incorrectly wrote the “sampling time length” as the “sampling period”, which caused misunderstanding. The sampling rate is 10kHz, which is equal to 0.1ms, and then we collected data for 1000 cycles at a time, with a total time of 100ms. We put these parameters in the table of the acquisition system to ease readability.

3. Comment:

“The data was collected 200 times for each class, but there is no mention on how different is each run from the other. It would be helpful if there are plots to illustrate the distribution of noise/nonlinearities/disturbances that might make each run unique, since the manuscript also stresses on the strong presence of nonlinearities in the GIS system.”
Response:

During the experiment, the sequence of data collection is normal case, isolation switch fault, looseness of flange screw and looseness of stone bolt. After each fault experiment, we restored the normal working condition, and then collected data for comparison with the previous normal working condition data, and used OCSVM to judge whether it is normal, so the experimental process is very complex.

The vibration signal of GIS system was measured by hammering method, and the natural frequency characteristics of the system were analyzed. Point B in figure (a) was knocked, and the time domain diagram and Fourier spectrum of vibration signal were measured at point a, as shown in figure (b).

(a)    Experiment and field testing

(b)    Inherent frequency analysis of vibration information

As can be seen from figure (b), under impact excitation, GIS vibration at test point A attenuates in a flared shape with time, and the system is characterized as a damped system. At the same time, it can be seen from the Fourier spectrum of the signal that the natural frequency of the system is mainly concentrated within 0.7~ 1.8kHz, and presents as the excitation mode of multiple frequency points.

In the three-phase GIS, it is assumed that the three-phase A, B and C are arranged in an equilateral triangle, and their top view and sectional view are shown in figure (c). The distance between wires is a, the length of conductors is l, and the three-phase current is symmetrical with frequency of 50Hz.

(c)     Mechanical analysis of three-phase GIS

It can be seen that the amplitude of the electric force of A phase conductor changes at a frequency of 2f (the period of the absolute sine function is twice the original function), and the frequency of the angle change is also 2f (the period of the tangent function is π). Therefore, the frequency of the electric force of A phase conductor is 100Hz. Similarly, the electric force frequency of B and C phases is also 100Hz.

The steady-state response of the linear damping system under 100Hz excitation is 100Hz, but the response to a 100Hz excitation in the experiment is not 100Hz in figure (d). Combined with the above analysis, we can see that the GIS system has a strong non-linearity.

(d)    Vibration signal and its spectrum in different working conditions

4. Comment:

“There is no indication on the intensity/details of the faults induced, i.e. faults with large or small magnitudes, and how many screws and nuts have to be loosened in order for the fault to be significant?”
Response:

In this paper, several typical failure conditions are selected for study, including looseness of stone bolt, looseness of flange screw and isolation switch fault.

Because the definition of fault degree is difficult, we only simulate the cases with severe fault degree. For example, bolt is completely loose, with almost no fixation and the isolation switch is the state when the contact finger just touches.

Considering the safety and feasibility of the experiments, we made the following choices when setting up the fault:

When setting looseness of stone bolt, considering each GIS shell with four stone bolt, loose three anchor bolt may cause the system structure is not stable, so chose to set up two bolt looseness on behalf of the looseness of stone bolt fault.

When setting the looseness of flange screw, considering that the flange is the connecting mechanism with 12 bolts, several bolts loosening may cause the structure to disintegrate, and three bolts loosening is already a serious failure.

When setting the isolation switch fault, because the contact length of the fingers of the three-phase isolation switch is dispersive, there must be a phase isolation switch that falls off first, and the phase that falls off first is selected to set the fault.

According to the research in this paper, it can be shown that the fault diagnosis method based on vibration signal analysis is feasible. On this basis, different degrees of failure can be studied in the follow-up work, and how many screws and nuts having to be in order for the fault to be significant can also be studied.

5. Comment:

“Please rectify Equation (7).”
Response:

Thank you for the input and Equation (7) has been rectified as:

6. Comment:

“Is that a symbol of a bird appearing before $p$ on Line 253?”
Response:

It is a symbol of ω in figure(e) not a bird in figure(f). We are sorry for the garbled code problem, which may be caused by different versions of the software.

(e)      

(f)      

7. Comment:

“Line 302: Any reason behind the choice of the values?”
Response:

Rth and Nth are very important, which determine the number of characteristic frequencies.Too many features can lead to feature redundancy, and too few features can lead to insufficient diagnostic information.

By observing the results of the coherence between signals, we can find that there are too many potential feature frequency points when Rth is 0.1 in figure (g). In this case, the value of Nth is evaluated and the characteristic frequency is observed

(g)   Rth=0.1

When Nth is 0.1, as shown in figure (h), the characteristic frequency points of each working condition are 500; When Nth is 0.9, as shown in figure (i), the number of characteristic frequency points in each working condition is more than half of the frequency points. It can be seen that the significance of feature screening has been little or even completely lost.

(h)    Rth=0.1 and Nth=0.1

(i)      Rth=0.1 and Nth=0.9

When Rth is 0.9, the number of potential characteristic frequency points is greatly reduced in figure (j), and the role of the first feature selection is very obvious. When Nth is equal to 0.1 and 0.9 respectively, the characteristic curves are shown in figure (k) and figure (l). Compared with the value of Rth being 0.1, the number of characteristic frequency points has been greatly reduced, but in the case of Nth being 0.1, the number of frequency points exceeds 100, and even reaches 200 in the looseness of flange screw fault.

 (j)      Rth=0.9

(k)    Rth=0.9 and Nth=0.1

(l)      Rth=0.9 and Nth=0.9

When Rth is 1, the number of potential characteristic frequency points is 0. According to the above analysis, too small Rth will lead to too many potential feature frequency points and lose the significance of feature screening, so we choose 0.5 as the lower limit of the value interval of Rth. Meanwhile, too large Rth cannot solve the influence of dispersion between signals, so we choose 0.95 as the upper limit of the value interval of Rth.

At the same time, we can get the conclusion about the value of Nth. Too small Nth cannot ensure the consistency of vibration signals in the characteristic spectrum under the same working condition, and too large Nth may cause the loss of characteristic frequency points and the decline of diagnostic accuracy. The selected characteristic frequency point must be able to represent the characteristics of most signals in such working conditions, so 0.6 is selected as the lower limit value of Nth. In consideration of signal dispersion, 0.95 is selected as the upper limit of the value interval of Nth.

8. Comment:

“Since the manuscript touches on 1-SVM and the detection of unknown faults, it would also be good to compare to recently published reports that address similar issues:

      — “Combining model-based diagnosis and data-driven anomaly classifiers for fault isolation”, Control Engineering Practice, 2018.

      — “A combined diagnosis system design using model-based and data-driven methods”, 2016 3rd Conference on Control and Fault-Tolerant Systems (SysTol), 2016.”
Response:

We have carefully read the literature you recommended and found it beneficial. A hybrid diagnosis system combining model—based diagnosis and data—driven anomaly classifiers for fault isolation was programs and its feasibility was verified in the internal combustion engine, which provides a new idea for the research of fault diagnosis. Similar to the OCSVM diagnosis of unknown faults in the paper, the recommended literature also used OCSVM to diagnose complex faults, providing new ideas for our future research.

And the statements of “What's more, it can judge the unknown faults, and the feature has been applied in the field of fault diagnosis. For example, system combining model-based diagnosis and data-driven anomaly classifiers for fault isolation used OCSVM to identify unknown faults and the validity of the method is verified in internal combustion engine” were added in the line 70-73.

9. Comment:

“There are some grammatical errors throughout the manuscript. Please revise.”
Response:

Thanks for your comments. We have carefully reviewed the whole paper, proofread and revised some words and grammar. The revised portions are marked in red in the manuscript.

We earnestly appreciate the reviewers’ hard work, and hope that the revised manuscript meets with your approval. We have tried our best to improve the manuscript. The changes made will not influence the content or framework of the paper.

Once again, we thank you for your comments and suggestions.

Round  2

Reviewer 1 Report

-

Author Response

Thank you for your letter and for the reviewers’ comments on our manuscript titled “Frequency Feature Learning from Vibration Information of GIS for Mechanical Fault Detection” (ID: sensors-470115).

Your comments have been very helpful for revising and improving our paper. They are also important and valuable guidelines significant to our research. We have studied the comments carefully and have made the relevant corrections, which we hope meet with your approval. The revised portions are marked in red in the manuscript(revision version v2). 

We earnestly appreciate the reviewers’ hard work, and hope that the revised manuscript meets with your approval. We have tried our best to improve the manuscript. The changes made will not influence the content or framework of the paper.

Once again, we thank you for your comments and suggestions.

Reviewer 2 Report

Dear Author

I am happy with the improvements made to the paper.

The grammatical and typographical errors should be revised.

Author Response

(The authors gave the same response as above.)
